# Extraction of Magnesium and Nickel from Nickel-Rich Serpentine with Sulfation Roasting and Water Leaching

**Xiyun Yang, Lingyu Gao \*, Yulou Wu, Yingli Chen and Lirong Tong**

School of Metallurgy and Environment, Central South University, Changsha 410083, China; yxy7412@csu.edu.cn (X.Y.); 18206795474@163.com (Y.W.); c1094047565@163.com (Y.C.); 18972759159@163.com (L.T.)
\* Correspondence: gly373210069@126.com

**Abstract:** Magnesium and nickel were recovered from nickel-rich serpentine through sulfation roasting and water leaching. The factors affecting the extraction percentages of Mg and Ni were discussed. Under the conditions of the ratio of acid to ore of 0.8:1 and roasting temperature of 650 °C for 120 min, 91.6% of Mg and 88.7% of Ni but only 4.8% of Fe were extracted. The roasting kinetics of Mg and Ni were investigated. The results showed that the roasting stage was governed by internal diffusion in the temperature range of 350–650 °C, and the activation energy of nickel and magnesium were different in the time ranges of 0–30 min and 60–120 min, with 17.45 kJ·mol$^{-1}$ (0–30 min) and 14.12 kJ·mol$^{-1}$ (60–120 min) for magnesium and 15.48 kJ·mol$^{-1}$ (0–30 min) and 12.46 kJ·mol$^{-1}$ (60–120 min) for nickel. The kinetic equations were obtained.

**Keywords:** serpentine; sulfation roasting; magnesium; nickel; kinetics

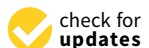



## 1. Introduction

Serpentine belongs to layered silicate minerals and exhibits a stable structure with the chemical formula [(Mg,Fe)$_3$Si$_2$O$_5$(OH)$_4$] [1,2]. It is a group of antigorite, chrysotile and lizardite and is composed of silica tetrahedra and magnesium hydroxide octahedra at a molar ratio of 1:1 [3,4]. Except for magnesium, silicon, and iron, serpentine contains a certain amount of nickel and other valuable elements (such as cobalt), Therefore, it is considered a potential nickel source [5,6].

Due to the similar ionic radius, Ni$^{2+}$ usually replaces Mg$^{2+}$ in serpentine and mainly exists as silicate accompanied by a small part of sulfide. Compared with ordinary nickel sulfide, serpentine has a lower nickel grade (0.2%) and higher magnesium oxide and silicate content [7,8]. It is difficult to find an economic method to recover nickel from serpentine. Recently, many researchers have concentrated on the recovery of magnesium, nickel and other metals from serpentine through hydrometallurgical routes. Different leaching agents such as sulfuric acid, hydrochloric acid and organic acids have been systematically investigated [9–12]. Pretreatment methods such as mechanical activation, thermal activation and calcination have been used to enhance the leaching process [13–15]. All cases suffer from high acid consumption, long leaching time and the need to remove various impurities such as iron, aluminum and chromium. Removing iron leads to a loss of nickel and magnesium due to the coprecipitation and adsorption of iron-containing residue. Therefore, it is very necessary to develop a technically and economically feasible process to extract valuable metals from serpentine.

Sulfation roasting is an effective method for processing complex and lean ores [16,17]. This process is carried out to convert metal oxides or sulfides into metal sulfates that are easily leached with water [18,19]. Compared with other pyrometallurgical methods, with low roasting temperature (200–700 °C), short process flow, high metal recovery and high reaction selectivity, it is considered a promising method in the field of nonferrous metal processing and has been extensively used to recycle some metals such as lithium, zinc,

aluminum and rare-earth elements from raw materials [20–23]. However, reports on the recovery of the metals magnesium and nickel from serpentine are rather scarce.

In this work, sulfation roasting and water leaching methods were used to recover magnesium and nickel in nickel-rich serpentine from Inner Mongolia. The selective separation of metals is achieved and the difficulties in the process of iron removal are avoided. The impacts of concentrated sulfuric acid amounts and roasting temperature as well as roasting time for the extraction efficiency of Mg, Ni and Fe were studied in detail. In addition, the reaction mechanism during the roasting period was investigated, and the kinetics for the roasting reaction phase was analyzed using the shrinking core model.

## 2. Experimental

### 2.1. Materials

The serpentine mineral was from Jiujingzi, Arukorqinqi, Inner Mongolia. The main components are shown in Table 1. Figure 1 reveals that the main mineral phases include antigorite [$Mg_3Si_2O_5(OH)_4$], nepouite [$(Ni,Mg)_3Si_2O_5(OH)_4$], lizardite [$(Mg,Fe)_3Si_2O_5(OH)_4$], magnetite ($Fe_3O_4$) and hematite ($Fe_2O_3$). The analytical-grade sulfuric acid was provided by Longxi Chemical Reagent Co., Ltd., Shantou, China. All the aqueous solutions were obtained with pure water.

**Table 1.** Main chemical components of serpentine.

| Component | MgO | $Al_2O_3$ | $Fe_2O_3$ | $Co_3O_4$ | NiO | $Cr_2O_3$ | MnO | $SiO_2$ |
|---|---|---|---|---|---|---|---|---|
| Content (wt.%) | 36.08 | 0.58 | 12.85 | 0.05 | 1.24 | 0.19 | 0.15 | 38.05 |

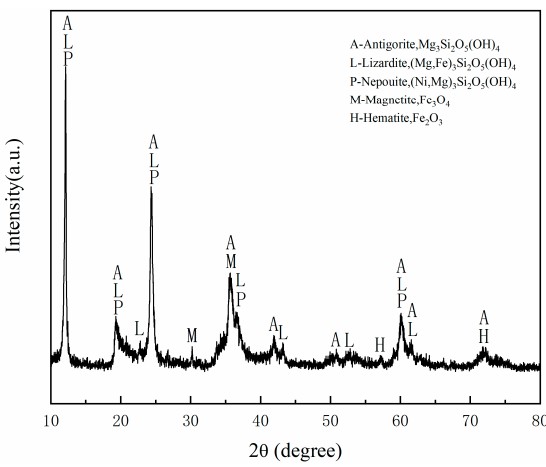

**Figure 1.** XRD pattern of serpentine sample.

### 2.2. Experimental Procedure

Firstly, the serpentine mineral was mixed evenly with concentrated sulfuric acid in a corundum crucible. Next, the mixture was placed in a muffle oven and roasted at the set temperature. After a certain while, the roasted samples were leached with water in a three-flask glass reactor equipped with a reflux condenser, and the stirring speed was kept at 400 rpm. The leaching conditions were a temperature of 60 °C, liquid/solid ratio of 8:1, and reaction time of 60 min. Following completion of the leaching procedure, the suspension was filtered to separate the solid from the liquid. The solid residue was washed and dried at 100 °C. The concentrations of metal ions in solution and filter residue were examined by ICP-OES. From the above analysis, the extraction rate of various metals can be calculated from Equation (1). The typical process flowchart for sulfation roasting and water leaching is shown in Figure 2.

$$\eta_a = \eta_b = \frac{cv}{m_0 w} \tag{1}$$

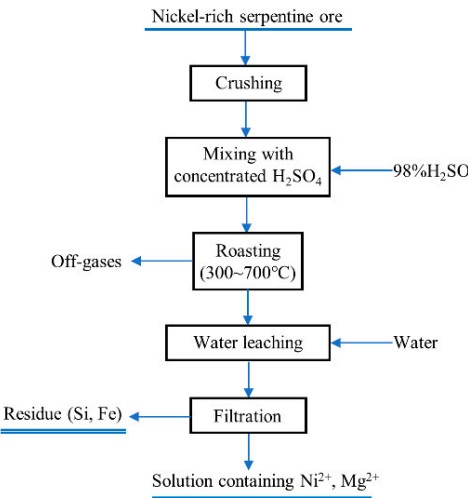

**Figure 2.** Typical process flowchart for sulfation roasting and water leaching.

Where $\eta_a$ is the extraction rate of the metal elements, $\eta_b$ is the sulfation rate of the metal elements, $c$ (g/L) is the concentration of metal ions in the filtrate, $v$ (L) is the volume of the filtrate, $w$ (%) is the content of metal elements in the ore, and $m_0$ (g) is the mass of the ore powders.

### 2.3. Characterization

XRD (Rigaku-SmartLab, Cu/Ka, Tokyo, Japan) was used to identify the phase components of samples, a 2θ range between 10 and 80 deg was scanned, and the scan speed was 10 deg/min. Thermogravimetry–differential scanning calorimetry (TA, SDT-Q600, New Castle, DE, USA) was used to reveal the thermal behavior of the mixture of serpentine and concentrated sulfuric acid in an air environment, and the temperature was raised to 1010 °C at 5 °C/min. An ICP-OES instrument (Shimadzu, ICPS-7210, Tokyo, Japan) was employed to determine the concentrations of Mg, Ni and Fe in the leachate. SEM (JEOL, JSM-6360LV, Tokyo, Japan) was employed to analyze the morphology of the samples.

## 3. Results and Discussion

### 3.1. Thermal Analysis in Roasting Stage

Figure 3 presents the thermogravimetry–differential scanning calorimetry (TG-DTA) curve for the mixture of serpentine with concentrated sulfuric acid (ratio of 0.8 mL/g). During the roasting process, a series of decomposition reactions occurred, and the thermal decomposition temperatures for some sulfates are shown in Table 2. Throughout the whole heating process, there were four obvious weight losses, and one exothermic peak and three endothermic peaks appeared on the corresponding DTA curves at 183 °C, 265 °C, 591 °C and 989 °C, respectively. The first intense exothermic peak was related to the chemical reaction of the mixture. The weight loss below 340 °C resulted from excess sulfuric acid, free water, and loss of crystal water in sulfates such as $FeSO_4(H_2O)$, $NiSO_4(H_2O)$, and $MgSO_4(H_2O)$. Furthermore, $FeSO_4(H_2O)$ also oxidizes into $FeOHSO_4$ during this temperature range. During 525~630 °C, the weight loss in the temperature was attributed to the decomposition of $Fe_2(SO_4)_3$, $Fe_2O(SO_4)_2$ and $Al_2(SO_4)_3$. With the temperature increased to 890–1000 °C, magnesium sulfate began to decompose as indicated by the obvious weight loss.

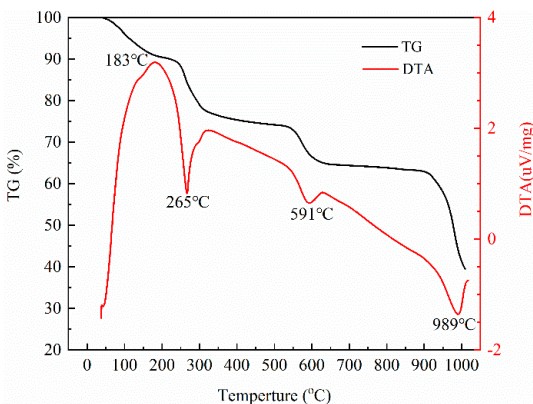

**Figure 3.** Thermogravimetric-differential thermal analysis of roasting process.

**Table 2.** Thermal decomposition temperature for some sulfates.

| Sulfates | Decomposition Temperature | Reference |
|----------|---------------------------|-----------|
| $Al_2(SO_4)_3$ | 524 | Tagawa [24] |
| $FeSO_4$ | 550 | Onal [25] |
| $Fe_2(SO_4)_3$ | 545 | Kolta et al. [26] |
| $MgSO_4$ | 890 | Pekka et al. [27] |

### *3.2. Sulfation Roasting Process*

#### 3.2.1. Effect of Sulfuric Acid Amount

Among the factors impacting the roasting stage, the first consideration should be the amount of sulfuric acid. The conditions of 650 °C and 120 min were kept constant, and several sets of roasting experiments were conducted by altering the ratio of sulfuric acid amount to serpentine (0.5–1 mL/g). As can be seen from Figure 4a, the extraction rate of Mg, Ni, and Fe increased with the ratio of acid to serpentine. When the ratio of acid to serpentine increased to 0.8 mL/g, the extraction of Mg and Ni reached 91.6% and 88.7%, respectively. After further increasing the sulfuric acid dosage, the extraction of Mg and Ni showed no obvious change, but that of iron improved. The ratio was controlled as 0.8:1.

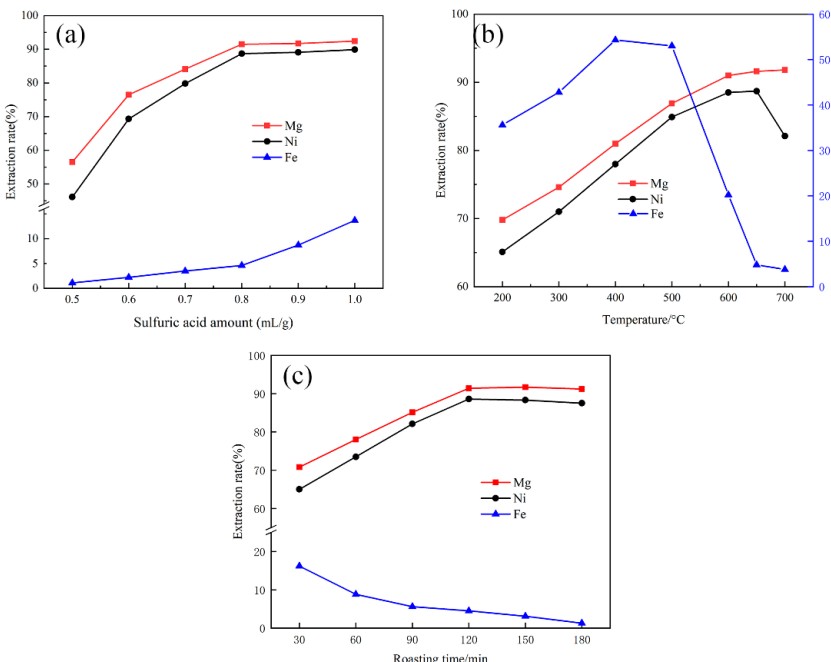

**Figure 4.** The effects of roasting parameters: (**a**) sulfuric acid amount, (**b**) roasting temperature, and (**c**) roasting time.

### 3.2.2. Effect of Roasting Temperature

The effect of roasting temperature on extraction rate of different elements at an acid to ore ratio of 0.8 mL/g with 120 min roasting time is presented in Figure 4b. When the roasting temperature was within the range of 200–650 °C, the extraction of magnesium and nickel increased with the temperature. For roasting temperatures beyond 650 °C, nickel extraction rate showed a downward trend. Moreover, the extraction rate of Fe began to decrease rapidly as the roasting temperature increased beyond 500 °C. The lower extraction rate at a higher temperature was possibly due to the decomposition of iron or nickel sulfate into their own oxides. The optimal roasting temperature was set as 650 °C.

### 3.2.3. Effect of Roasting Time

The effect of roasting time on the extraction rate of Mg, Ni and Fe was investigated under the optimum conditions: acid to ore ratio, 0.8 mL/g; temperature, 650 °C. The obtained results are shown in Figure 4c. The extraction of Mg and Ni increased rapidly when the roasting time was extended from 30 to 120 min. Further increasing the roasting time had no significant effect on the extraction rate of Mg and Ni after 120 min. In addition, the longer time favored the decomposition of iron sulfates into oxides and led to a lower extraction rate. Hence, 120 min was the desirable roasting time.

### 3.3. Roasting Mineralogy Analysis

Figure 5 shows the XRD patterns of the calcine obtained at different roasting temperatures. From Figure 5, it can be seen that the main phases of the calcine were magnesium sulfate, nickel sulfate, iron sulfate and iron sulfate oxide. During roasting, the reactions between serpentine and concentrated sulfuric acid can be expressed as the following [28–30]:

$$Mg_3Si_2O_5(OH)_4 + 3H_2SO_4 \rightarrow 3MgSO_4 + 2SiO_2 + 5H_2O \tag{2}$$

$$(Mg, Fe)_3Si_2O_5(OH)_4 + 6H_2SO_4 + 3/2O_2 \rightarrow 3MgSO_4 + 3FeSO_4 + 2SiO_2 + 8H_2O \tag{3}$$

$$(Ni, Mg)_3Si_2O_5(OH)_4 + 6H_2SO_4 + 3/2O_2 \rightarrow 3MgSO_4 + 3NiSO_4 + 2SiO_2 + 8H_2O \tag{4}$$

$$Fe_3O_4 + 4H_2SO_4 \rightarrow Fe_2(SO_4)_3 + FeSO_4 + 4H_2O \tag{5}$$

$$Fe_2O_3 + 3H_2SO_4 \rightarrow Fe_2(SO_4)_3 + 3H_2O \tag{6}$$

When the temperature increased from 300 °C to 400 °C, the peak intensity of magnesium and iron sulfate gradually increased. In the temperature range of 400~500 °C, the peaks of iron sulfate oxides $Fe(OH)SO_4$ and $Fe_2O(SO_4)_2$ appeared according to the following equations:

$$2FeSO_4 \cdot H_2O + 1/2\,O_2 \xrightarrow{300\sim500\,°C} 2FeOHSO_4 + H_2O \tag{7}$$

$$2FeOHSO_4 \xrightarrow{300\sim500\,°C} Fe_2(SO_4)_2 + H_2O \tag{8}$$

$$2FeSO_4 + 1/2\,O_2 \xrightarrow{400\sim500\,°C} Fe_2(SO_4)_2 \tag{9}$$

The peak intensity of iron sulfate and iron sulfate oxides gradually decreased at roasting temperatures above 500 °C, and the characteristic peaks of iron oxide appeared. The reason for this was because of decomposition reaction of $Fe_2O(SO_4)_2$ and $Fe_2(SO_4)_3$ occurred; the decomposition reaction equations can be written as [25,29]:

$$Fe_2O(SO_4)_2 \xrightarrow{500\,°C\sim} Fe_2O_3 + 2SO_3 \tag{10}$$

$$Fe_2(SO_4)_3 \xrightarrow{500\,°C\sim} Fe_2O_3 + 3SO_3 \tag{11}$$

These responses indicated that ferric sulfate began to decompose at 500 °C and totally transformed into iron oxide at 650 °C. The calcine obtained at 650 °C was composed of

magnesium sulfate ($MgSO_4$), nickel sulfate ($NiSO_4$), and iron oxide ($Fe_2O_3$). When the temperature rises above 700 °C, nickel sulfate may decompose. Combining the XRD results with the TG-DTA results, the phase transformation of metals during sulfation roasting and water leaching is shown in Figure 6.

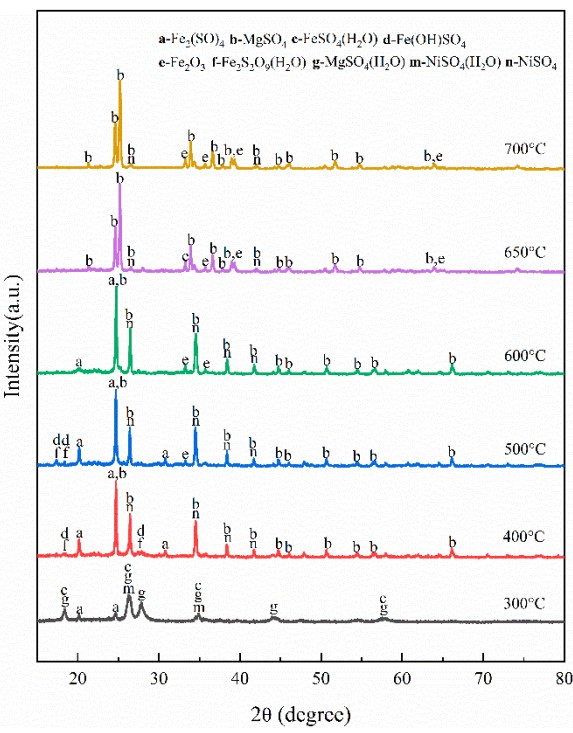

**Figure 5.** XRD patterns of calcine at different roasting temperatures.

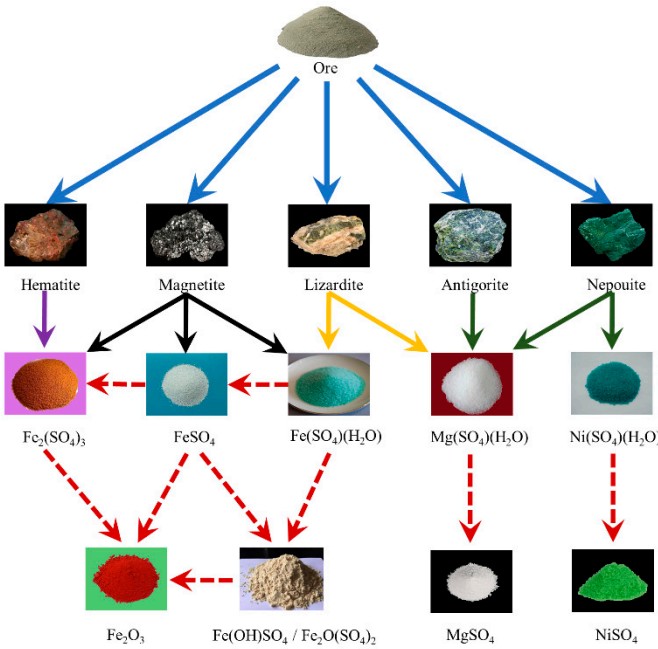

**Figure 6.** The phase transformation of metals during sulfation roasting and water leaching.

Figure 7a–d shows the SEM and EDS images of the serpentine mineral and water leaching residue. The serpentine sample exhibited an irregular shape and a wide size distribution. The residue particles were smaller than the particles of the original serpentine sample. The change in particle size might be due to the destruction of the silicate structure

of serpentine after roasting and water leaching. Moreover, as can be seen from the EDS data, the main elements of the surface of the residue were Si, Fe and O, while Mg and Ni were hardly detected. This indicates that most of the Mg and Ni in the roasting products had been leached. The SEM also shows that the $SiO_2$ particles in the residue were lumpy, with rough surfaces and large size, while $Fe_2O_3$ particles were agglomerated, and the particles were smaller than $SiO_2$.

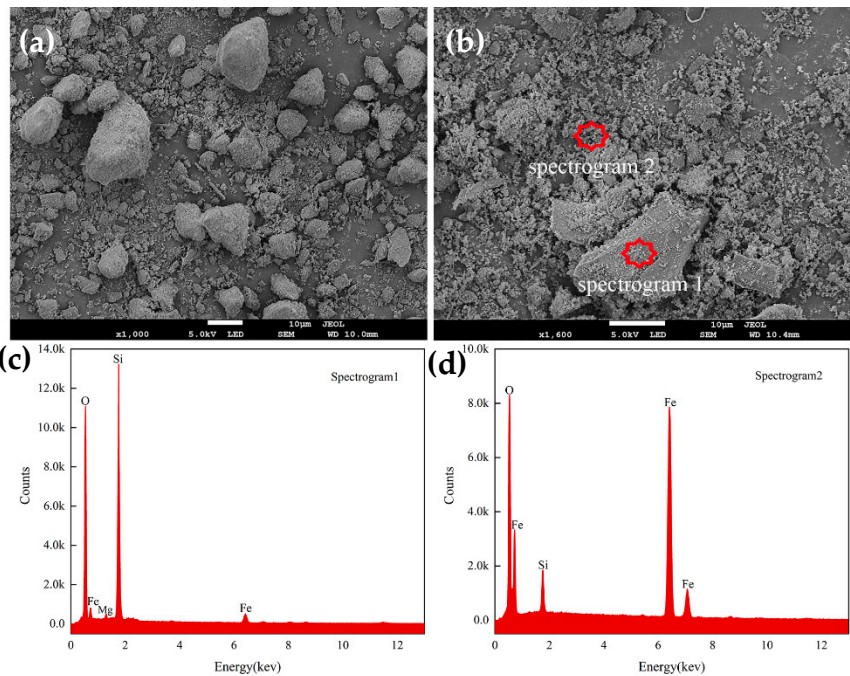

**Figure 7.** (**a**) SEM images of the serpentine sample. (**b**) SEM images of the water-leaching residue. (**c**,**d**) EDS data of the water-leaching residue.

### 3.4. Roasting Kinetics Study

The surface area of the serpentine particles decreased as the reaction proceeded. As shown in Figure 8a,b, the shrinking core model could be used to study the sulfation roasting stage by considering the reaction particles as spherical during 350–650 °C. The change in the sulfation rate for magnesium and nickel with time was divided into two obvious periods. One was between 0–30 min and the other was between 60–120 min.

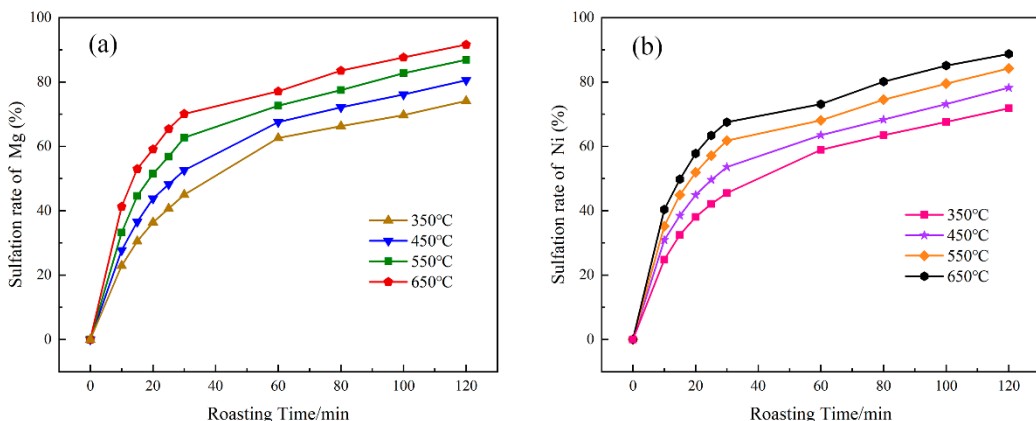

**Figure 8.** Relationship between the sulfation rate and roasting time at different temperatures, (**a**): magnesium; (**b**): nickel.

The simplified kinetic equations can be expressed as follows [31–34]:

$$x = k_f t \tag{12}$$

$$1 - (1 - x)^{1/3} = k_r t \tag{13}$$

$$1 - 2/3x - (1 - x)^{2/3} = k_d t \tag{14}$$

Where $k_f$, $k_r$ and $k_d$ are rate constants for the constant of external diffusion, chemical reaction-controlled process and internal diffusion, respectively. $x$ is the sulfation rate of magnesium or nickel, and t is the reaction time. The data in Figure 8a,b were fitted with Equations (12)–(14). It was found that the correlation coefficient of Equation (14) was higher than those of Equations (12) and (13). The plots of the left-hand side of Equation (14) versus time for magnesium and nickel under several temperatures are presented in Figure 9. It was observed that each plot can be fitted using two straight lines, meaning that the kinetics was divided into two stages. Activation energy ($E_a$) was obtained according to the following Arrhenius equation:

$$\ln k_d = \ln A - E_a / (RT) \tag{15}$$

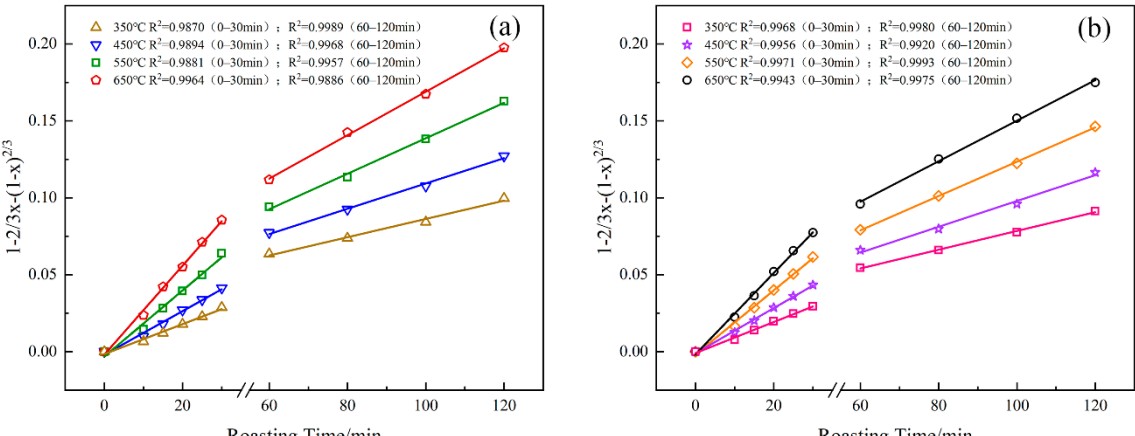

**Figure 9.** Relationship between $1 - (2/3)x - (1 - x)^{2/3}$ and t during roasting process. (**a**): magnesium; (**b**): nickel.

$A$ represents the frequency coefficient in the formula; $E_a$ denotes the activation energy. The relationships of $\ln k_d$ versus $1/T$ are given in Figure 10a,b.

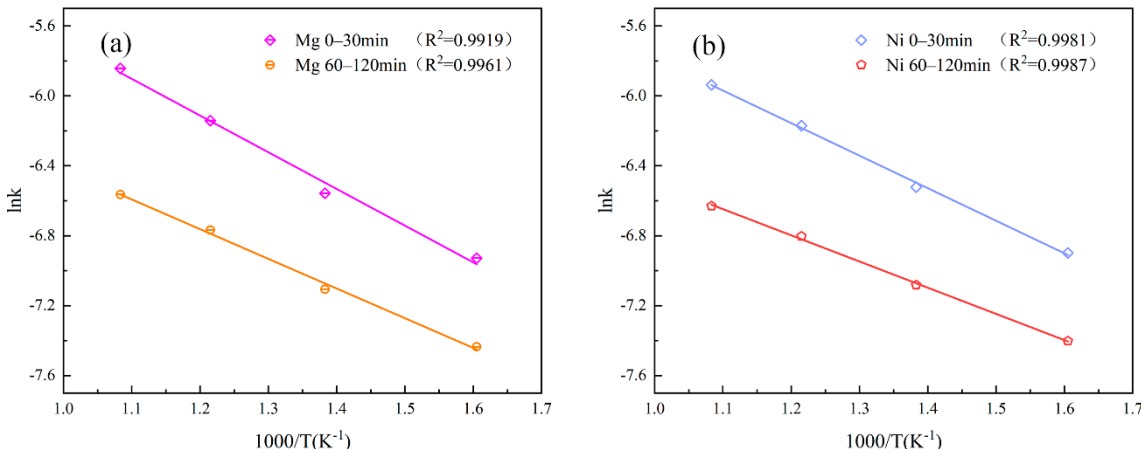

**Figure 10.** Relationship between $\ln k_d$ and $1/T$ during roasting process. (**a**): magnesium; (**b**): nickel.

The activation energy could be calculated from the slope of the lines in Figure 10a,b. The activation energy of magnesium roasting was 17.45 kJ·mol$^{-1}$ (0–30 min) and 14.12 kJ·mol$^{-1}$

(60–120 min). The activation energy of nickel roasting was 15.48 kJ·mol$^{-1}$ (0–30 min) and 12.46 kJ·mol$^{-1}$ (60–120 min). The results demonstrated that the initial stage activation energy was greater than that of the later stage.

The reason for this was that, under controlled conditions of internal diffusion, the main factor affecting the reaction rate was the change of reactant concentration. Since in the initial stage of the reaction (0–30 min), the amount of sulfuric acid was sufficient, the reaction mainly occurred between liquid and solid, and the reaction rate was relatively fast. In the later stage of the reaction (60–120 min), due to insufficient amounts of sulfuric acid, the reaction occurred between liquid and solid as well as gas and solid, and the reaction rate was slower.

Finally, it could be confirmed from the activation energy ($E_a$) that the roasting stage was governed by the internal diffusion. Kinetic model equations in the sulfation roasting process are shown in Table 3.

**Table 3.** Kinetic model equations in the sulfation roasting process.

| Metals | Kinetic Equation | |
|---|---|---|
| | **0–30 min** | **60–120 min** |
| Mg | $1 - (2/3)x - (1 - x)2/3 = 0.0275t \exp[-17,450/(RT)]$ | $1 - (2/3)x - (1 - x)2/3 = 0.0089t \exp[-14,120/(RT)]$ |
| Ni | $1 - (2/3)x - (1 - x)2/3 = 0.0198t \exp[-15,480(RT)]$ | $1 - (2/3)x - (1 - x)2/3 = 0.0067t \exp[-12,460/(RT)]$ |

## 4. Conclusions

In this work, a selective sulfation roasting process for nickel-rich serpentine using concentrated sulfuric acid followed by water leaching was proposed. The major conclusions obtained in this study are as follows:

(1) Based on TG/DTA and XRD, the sulfation roasting process was studied, and the transformation mechanism of the main mineral phases of nickel-rich serpentine during roasting was revealed. MgO, NiO, Fe$_2$O$_3$, and FeO in serpentine were converted into their corresponding sulfates during the sulfation roasting stage. With the increase in the roasting temperature to about 500 °C, the iron sulfate began to decompose to form Fe (III) oxide, and the increase in roasting temperature reduced the dissolution of iron. Silicon was present in the serpentine without chemical reaction.

(2) During sulfation roasting, the extraction rate of magnesium and nickel were affected by the amount of sulfuric acid, roasting temperature, and roasting time. The optimal sulfation roasting conditions were: acid to ore ratio of 0.8 mL/g; roasting temperature of 650 °C; roasting time of 120 min. Employing these experimental conditions, the extraction of Mg and Ni can achieve 91.6% and 88.7%, respectively, while the extraction of iron was less than 4.8%. Si was preserved as SiO$_2$ in the water-leaching residue.

(3) The kinetic analysis indicated that the sulfation roasting process follows the shrinking core model. The limiting step for the sulfation roasting stage was governed by internal diffusion at the temperature range of 350–650 °C. Kinetic equations during 0–30 min and 60–120 min were acquired, and the activation energy of magnesium and nickel in different periods was calculated from the Arrhenius equation.

The method was shown to be feasible for the extraction of nickel and magnesium from serpentine with relatively low production energy and costs, which provides useful information for the future development and utilization of nickel-rich serpentine.

**Author Contributions:** Conceptualization, X.Y. and L.G.; methodology, X.Y. and L.G.; investigation, Y.W.; data curation, L.G.; writing—original draft preparation, L.G.; writing—review and editing, X.Y. and L.G.; visualization, X.Y. and L.G.; supervision, Y.C. and L.T. All authors have read and agreed to the published version of the manuscript.

**Funding:** This research was funded by National Natural Science Foundation of China (Grant No. 51574286).

**Institutional Review Board Statement:** Not applicable.

**Informed Consent Statement:** Not applicable.

**Data Availability Statement:** Data sharing is not applicable.

**Conflicts of Interest:** The authors declare no conflict of interest.

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
