# Peer review of "Extraction of Magnesium and Nickel from Nickel-Rich Serpentine with Sulfation Roasting and Water Leaching"

_metals, doi:10.3390/met12020318_

Round 1
Reviewer 1 Report
- In the first paragraph of the Introduction, change the comma after elements, and before “considered as a potential…”, add “Therefore, it is”.
- In section 2.3, change the scan speed to deg/min from C/min.
- In 3.2.1, when considering the amount of sulfuric acid to use for the roasting process, did you consider calculating the theoretical minimum amount needed for the chemical reactions? In other words, given the chemical analysis of the serpentine listed in table 1 and the amount of serpentine mineral used under each experimental condition, the minimum amount of sulfuric acid needed for the full conversion may be calculated. Of course, during the calculation, if the equilibrium constant of each reaction is available (from literature or thermodynamics database), the excess driving force may also be determined.
- In section 3.2.2, “When the roasting temperature within the range of 200-650”, add a “is” after “temperature”.
- In section 3.2.2, for roasting temperature over 650, the recovery rate of Ni starts to decrease. It was not so obvious from the XRD in Figure 5 as the peaks of NiSO4 shown at 650C are still noticeable at 700 C. Maybe the intensity decreased a little bit. But it’s difficult to see. Have you looked up relevant literature/thermodynamics database on the decomposition temperature of NiSO4 under a similar partial pressure of SO2 and O2 as used in your experiments to support your conclusions here?
- For the kinetics analysis part, you concluded that the roasting process was controlled by diffusion of H2SO4 inside the serpentine mineral particles, and the activation energies of the diffusion coefficients were not constant. Normally, when this is true, there might be an abrupt structural change accompanied inside the mineral particles during the roasting process. This sudden change in structure either causes the particle to become denser (open pores blocked) or looser (new cracks opened). Has this been observed in your samples before and after 30 min? Have you also tried other fluid-solid reaction models other than the shrinking core model?
Author Response
Dear Reviewer:
Thank you for your comments concerning our manuscript entitled “Extraction of magnesium and nickel from nickel-rich serpentine with sulfation roasting and water leaching” (ID: metals-1564203). Those comments are all valuable and very helpful for revising and improving our paper, as well as the important guiding significance to our researches. We have studied comments carefully and have made correction which we hope meet with approval. Revised portion are marked in the paper. The main corrections in the paper and the responds to the reviewer's comments are as flowing:
- We have been filled with “Therefore, it is” before “considered as a potential…”
- We have changed the scan speed to deg/min from°C/min
- Before considering the sulfuric acid dosage experiments for the roasting process, we first calculated the theoretical minimum amount of sulfuric acid required based on the main chemical composition of serpentine in Table 1 and the chemical reaction equation. Then, we started to design the sulfuric acid dosage experiments and considered the effect of the excess factor. The theoretical minimum sulfuric acid dosage for the chemical reaction was calculated to be about 0.6 mL/g.
- We have added add a “is” after “temperature” in section 3.2.2
- Because the nickel grade in serpentine is low and may just reach the initial decomposition temperature of NiSO4, the peak intensity variation is not very obvious from the XRD in Fig. 5. In our experiments, we looked up the relevant literature/thermodynamics database for the decomposition temperature of NiSO4 under different atmospheres, and the NiSO4 decomposition temperature range is roughly between 640-750°C, and the relevant literature is shown below.
[1] Kolta G A, Askar M H. Thermal decomposition of some metal sulphates[J]. Thermochimica Acta, 1975, 11(1):65–72.
[2] Tagawa H. Thermal decomposition temperatures of metal sulfates[J]. Thermochimica Acta, 1984, 80(1):23-33.
[3] Nal M A R, Borra C R, Guo M, et al. Recycling of NdFeB Magnets Using Sulfation, Selective Roasting, and Water Leaching[J]. Journal of Sustainable Metallurgy, 2015, 1(3):199-215.
- As you said, the roasting process may be accompanied by structural changes inside the mineral particles. In this experiment, this was also observed in the samples before and after 30 minutes, where the particles became denser (open pores were blocked) and the roasted samples were harder and brittle.
The shrinking core model is used to describe the dissolution, leaching or reaction process of solid particles, which is more common in hydrometallurgy. In this experiment, we have not tried other fluid-solid reaction models yet, and we will work hard to learn and try other models for kinetic analysis in the future experiments.
Once again, thank you very much for your constructive comments and suggestions which would help us both in English and in depth to improve the quality of the paper.
Kind regards,
Corresponding author

Reviewer 2 Report
The paper presents a new way to recover magnesium and nickel from nickel-rich serpentine by using H2SO4 and roasting in air. Previous studies have pointed the possibilities to recover those metals by using ammonium chloride, hydrochloric acid, reduction roasting. Optimal conditions (acid to mineral ratio, roasting temperature and time) for sulfating roasting have been determined, as well as the optimal conditions for the calcine leaching, although the latter briefly - only in the conclusion . The proposed methodology and obtained conditions can be very useful tool for specialists dealing with nickel-rich serpentine.
The proposed equations for the reactions between serpentine and concentrated sulfuric acid during roasting (based on the TG/DTA and XRD) shed light on the mechanism of transformations of the serpentine mineral phases during roasting.
The authors have attempted to fit their data to some simplified kinetic equations.
However, it is not clear whether the data are for roasting kinetics (as it is written - 3.4. Roasting kinetics study) or the data are for extraction (The change of extraction rate for magnesium...- page 7, lines 4,5 from the bottom).
It is written on lines 5,6 from the top, page 8 "x is the extraction rate of magnesium or nickel" and further on page 11, lines 12-14 from the top "it could be confirmed from the activation energy (Ea) that the roasting stage was governed by the internal diffusion".
In addition, if the ''extraction rate'', that is not defined in the paper and have to be, means the % of the metal extracted with the respect of the available in the raw material (as implied by the fig. 8) then that overall rate includes also the leaching kinetics.
The discussion related to the kinetics and activation energy has to be clarified.
In my opinion, dividing the process into two time domains and finding 2 values for the activating energy for each of the metals is artificial. The values found are very close and simply correspond to a diffusion-controlled process.
The conclusion is written vaguely, with technical errors and inaccuracies in terminology. It should be edited. More comments and proposals are available in the attached file.

Author Response
Dear Reviewer:
First of all, we thank you for your affirmation of our research. We have studied reviewers’ comments carefully. In addition, according to the reviewers’ detailed suggestions, we have made a careful revision on the original manuscript. All revised portions are marked in red in the revised manuscript which we would like to submit for your kind consideration. Finally, regarding the discussion in the kinetics and conclusions section we make the following explanations:
Roasting and leaching are two processes. We use the extraction rate to define the roasting process, which is easy to be confused with the leaching process, which is inappropriate. We apologize for the confusion it brings to your review. We have redefined the concept of sulfation rate in the experimental procedure to express the degree of sulfation roasting. In addition, we have revised the content of the conclusion section, enriched the content, and made a summary and analysis. For a more detailed line-by-line response, please see in the attachment.
Once again, thank you very much for your constructive comments and suggestions which would help us both in English and in depth to improve the quality of the paper. Last but not least, on the occasion of the Chinese Lunar New Year, I wish you a happy and prosperous Year of the Tiger!
Kind regards,
Corresponding author

Reviewer 3 Report
Extraction of magnesium and nickel from nickel-rich serpentine with
sulfation roasting and water leaching is very important paper in extractive metallurgy. Some improvement is required.
Introduction: . The results showed that the roasting stage was governed
by internal diffusion, and the activation energy of nickel and magnesium were different in the time range of 0-30 minutes and 60-120 minutes, with 17.45 kJ⋅mol-1 (0-30 min) and 14.12 kJ⋅mol-1 (60-120 min) for magnesium and 15.48 kJ⋅mol-1 (0-30 min) and 12.46 kJ⋅mol-1 (60-120 min) for nickel. (Please to add temperature interval for this temperature interval
Page 1: and other valuable elements (such as cobalt)
page 1: Compared with other pyrometallurgical methods, with low
roasting temperature (please to write this value)
PAge 3: at Figure 2 you can write Residue (Si, Fe)
Page 3(3.1): Please to write chemical equations regarding oxidation and decomposition reaction related to TGA/DTA analysis
Page 4: When the ratio of acid to serpentine increased to 0.8 mL/g, the extraction of Mg and Ni reached 91.6% and 88.7%, respectively. What is the extraction of iron under same conditions?
Page 4: Moreover, the extraction rate of Fe (which value?) begins to decreased for the roasting temperature over 500°C (what is the extraction value of iron?) Which tip of iron oxide shall be formed during decomposition?
Page 6: When the temperature rose above 700°C, nickel sulfate may decompose. I am not sure for this statement. Can you write chemical reaction for this decomposition,
Page 7, at Figure 8: Extraction of Mg (%)
Page 8: x is the extraction rate of magnesium or nickel (during roasting?)
Page 8: Please to write correct: (11), (12), (13)
Page 8: Conclusion: and silicon was present in the slag as SiO2. This is not correct. The slag is product after smelting process, but during sulfation roasting, a smelting process is missing. SiIlicon is present in serpentine, and no chemical reaction.
Author Response
Dear Reviewer:
Thank you for your comments concerning our manuscript entitled “Extraction of magnesium and nickel from nickel-rich serpentine with sulfation roasting and water leaching” (ID: metals-1564203). Those comments are all valuable and very helpful for revising and improving our paper, as well as the important guiding significance to our researches. We have studied comments carefully and have made correction which we hope meet with approval. Revised portion are marked in the paper. The main corrections in the paper and the responds to the reviewer's comments are as flowing:
- We have added temperature interval for the kinetics of nickel and magnesium during the roasting stage.
- We have added valuable elements (such as cobalt).
- We have added sulfation roasting temperature range.
- In Figure 2, we have written the residue (Si, Fe).
- The chemical equations for the oxidation and decomposition reactions related to TG/DTA analysis are shown below(Please check the attachment)
- When the ratio of acid to serpentine increased to 0.8 mL/g, the extraction of Mg and Ni reached 91.6% and 88.7%, respectively. The extraction rate of iron under the same conditions was 4.6%.
- In addition, when the roasting temperature exceeded 500°C, the extraction rate of iron began to decline rapidly (the extraction value of iron was 53%). When the temperature reached 650°C, the extraction rate of iron was 4.8%, and when the temperature reached 700°C, the decomposition of iron sulfate was the most complete and the extraction rate of iron was the lowest (3.8%), but the extraction rate of nickel also decreased at this time, so the best roasting temperature was 650°C under comprehensive consideration.
- The chemical reaction of nickel sulfate decomposition is as follows(Please check the attachment):
In our experiments, we looked up the relevant literature/thermodynamics database for the decomposition temperature of NiSO4 under different atmospheres, and the NiSO4 decomposition temperature range is roughly between 640-750°C, and the relevant literature is shown below.
[1] Kolta G A, Askar M H. Thermal decomposition of some metal sulphates[J]. Thermochimica Acta, 1975, 11(1):65–72.
[2] Tagawa H. Thermal decomposition temperatures of metal sulfates[J]. Thermochimica Acta, 1984, 80(1):23-33.
[3] Nal M A R, Borra C R, Guo M, et al. Recycling of NdFeB Magnets Using Sulfation, Selective Roasting, and Water Leaching[J]. Journal of Sustainable Metallurgy, 2015, 1(3):199-215.
- Thanks to you for your good comments. We have modified the Y axis title.
- Thank you for your careful discovery, we have corrected it, x is the sulfation rate of magnesium or nickel during roasting.
- Thanks for your concern. We have checked the equations.
- The conclusion section has been revised to say that silicon was present in serpentine without chemical reaction.
Once again, thank you very much for your constructive comments and suggestions which would help us both in English and in depth to improve the quality of the paper.
Kind regards,
Corresponding author

Reviewer 4 Report
I would like to thank the authors for their study. Here are the parts that need to be corrected in this study:
- First of all, it will be more understandable if the XRD curve is enlarged a little more.
- The conditions of water leaching should be specified.
- The English of Section 2.2 Experimental Procedure should be corrected
- References should be given for the equations given in the Roasting section.
- It appears as "Onal" in the source text number 25. It is given as "Oeal" in the references section. It should be corrected.
- In Figure 4b, the expression "Extraction of Mg and Ni" is used on the y-axis, but "Fe" is also present in the figure. This expression should be changed as "Extraction Rate" as in other figures.
- Conclusion part is weak. Only the performing tests are mentioned. Final results need to be given and analyzed in detail. Finally, only the Mg and Ni values taken into the solution are given. These two elements must be obtained by selective precipitation. I would like to know if this experiment has been done.
Author Response
Dear Reviewer:
Thank you for your comments concerning our manuscript entitled “Extraction of magnesium and nickel from nickel-rich serpentine with sulfation roasting and water leaching” (ID: metals-1564203). Those comments are all valuable and very helpful for revising and improving our paper, as well as the important guiding significance to our researches. We have studied comments carefully and have made correction which we hope meet with approval. Revised portion are marked in the paper. The main corrections in the paper and the responds to the reviewer's comments are as flowing:
- The XRD curve has been slightly enlarged.
- Thanks for your suggestion, we have specified the conditions of water leaching in the experimental procedure.
- We have corrected the English of the experimental procedure in Section 2.2.
- References have given for the equations given in the Roasting section.
- We have matched the author's name in the reference to "Onal" in text number 25.
- We have unified the expression of the y-axis title in Figure 4.
- First of all, we have revised the content of the conclusion section, enriched the content, and made a summary and analysis. In addition, in this study we focus on the investigation of the mineral phase transformation mechanism and kinetics of the roasting process, therefore, only the Mg and Ni values in solution are given at the end. As you said, these two elements must be obtained by selective precipitation, and we subsequently chose the sulfide precipitation method to recover nickel and the ammonia precipitation process for magnesium. In the following research we will continue to discuss.
Once again, thank you very much for your constructive comments and suggestions which would help us both in English and in depth to improve the quality of the paper.
Kind regards,
Corresponding author

Round 2
Reviewer 4 Report
The article is more understandable in its current form.